# Characteristic Chest Computed Tomography Findings for Birt–Hogg–Dube Syndrome Indicating Requirement for Genetic Evaluation

**DOI:** 10.3390/diagnostics13020198

**Published:** 2023-01-05

**Authors:** Yong Jun Choi, Chul Hwan Park, Hye Jung Park, Jae Min Shin, Tae Hoon Kim, Kyung-A Lee, Duk Hwan Moon, Sungsoo Lee, Sang Eun Lee, Min Kwang Byun

**Affiliations:** 1Pulmonary Division, Department of Internal Medicine, Gangnam Severance Hospital, Yonsei University College of Medicine, Seoul 06273, Republic of Korea; 2Department of Radiology and the Research Institute of Radiological Science, Gangnam Severance Hospital, Yonsei University College of Medicine, Seoul 06273, Republic of Korea; 3Department of Laboratory Medicine, Gangnam Severance Hospital, Yonsei University College of Medicine, Seoul 06273, Republic of Korea; 4Department of Thoracic Surgery, Yonsei University College of Medicine, Seoul 06273, Republic of Korea; 5Department of Dermatology, Yonsei University College of Medicine, Seoul 06273, Republic of Korea

**Keywords:** Birt–Hogg–Dube syndrome, cystic lung disease, chest computed tomography, *FLCN* mutation

## Abstract

Background: Chest computed tomography (CT) findings are important for identifying Birt–Hogg–Dube (BHD) syndrome. However, the predictive power of classical criteria for chest CT findings is weak. Here, we aimed to identify more specific chest CT findings necessitating genetic examination for *FLCN* gene mutations. Methods: From June 2016 to December 2017, we prospectively enrolled 21 patients with multiple bilateral and basally located lung cysts on chest CT with no other apparent cause, including cases with and without spontaneous primary pneumothorax. All enrolled patients underwent *FLCN* mutation testing for diagnosis confirmation. Results: BHD was diagnosed in 10 of 21 enrolled patients (47.6%). There were no differences in clinical features between the BHD and non-BHD groups. Maximal cyst diameter was significantly greater in the BHD group (mean ± standard deviation; 4.1 ± 1.1 cm) than in the non-BHD group (1.6 ± 0.9 cm; *p* < 0.001). Diversity in cyst size was observed in 100.0% of BHD cases and 18.2% of non-BHD cases (*p* = 0.001). Morphological diversity was observed in 100.0% of BHD cases and 54.6% of non-BHD cases (*p* = 0.054). Areas under the receiver operating characteristic curves for predicting *FLCN* gene mutations were 0.955 and 0.909 for maximal cyst diameter and diversity in size, respectively. The optimal cut-off value for maximal diameter *FLCN* mutations prediction was 2.1 cm (sensitivity: 99%; specificity: 82%). Conclusions: Reliable chest CT features suggesting the need for *FLCN* gene mutations screening include variations in cyst size and the presence of cysts > 2.1 cm in diameter, predominantly occurring in the bilateral basal lungs.

## 1. Introduction

First reported in 1977, Birt–Hogg–Dube syndrome (BHD) is a rare genetic disorder caused by mutations in the folliculin gene (*FLCN*) [1,2,3]. Although BHD incidence remains unknown, the estimated prevalence of BHD in the general population is approximately 1/200,000 [4,5,6]. Recently, BHD is increasingly been diagnosed [6,7] which may be related to advancements in diagnostic technologies and increased awareness of the disease.

The clinical presentations of BHD include cutaneous manifestations (fibrofolliculoma, acrochordons; 84%), pulmonary cysts (70–85%), spontaneous recurrent pneumothorax (~25%), and renal cell carcinoma (19–35%) [3,8,9,10]. As these clinical manifestations can differ according to age group and patient characteristics, BHD remains difficult to diagnose, likely resulting in many cases of missed diagnosis [7,11]. Especially in Asia, skin manifestations and renal cell carcinoma are not predominant features of BHD, Therefore, in order to diagnose a disease that does not show symptoms or signs at an early stage, highlighting the importance of chest computed tomography (CT) findings in suspected cases [12].

In 2009, Menko et al. proposed modified diagnostic criteria for BHD based on the detection of DNA mutations in *FLCN,* citing chest CT features of BHD as an important minor criterion. Such features include multiple bilateral lung cysts in the basal region with no other apparent cause, presenting with or without spontaneous primary pneumothorax [7]. Using these widely accepted criteria, our previous study identified 17 patients with characteristic chest CT findings, and only six patients were finally diagnosed with BHD (35.3%) [11]. Therefore, characteristic chest CT findings beyond the classical criteria are required to improve the predictive power of CT for BHD. In this study, we aimed to identify which chest CT findings necessitate genetic examination for *FLCN* mutations.

## 2. Methods

### 2.1. Patients

From June 2016 to December 2017, we screened all patients who underwent chest CT and prospectively enrolled patients with multiple bilateral and basally located lung cysts on chest CT, which are minor criteria for the diagnosis of BHD in previous studies [7,13]. Patients exhibiting cysts with other apparent causes were excluded such as chronic obstructive pulmonary disease, lymphangioleiomyomatosis, pulmonary Langerhans cell histiocytosis, and lymphocytic interstitial pneumonia, although cases both with and without spontaneous primary pneumothorax were included. All enrolled patients underwent screening for *FLCN* gene mutations to confirm their diagnosis (see Appendix A, Figure A1).

The following clinical data were collected for all enrolled patients: age, sex, smoking status, comorbidities, urinalysis, pulmonary function test (PFT) results, personal and family history of spontaneous pneumothorax, presence of skin lesions (evaluated by dermatologists), and renal lesions (evaluated by radiologists).

### 2.2. CT protocol and Analysis

Chest CT was performed using either a 64-slice MDCT scanner (Somatom Sensation 64; Siemens Medical Solutions, Erlangen, Germany) or a 128-slice MDCT scanner (Somatom Sensation AS+; Siemens Medical Solutions, Erlangen, Germany or Ingenuity Core 128, Philips Healthcare, Cleveland, OH, USA). After acquiring the scout image to determine the field-of-view, conventional CT scanning was performed with a 1–3 mm reconstruction interval in the mediastinal window setting. The exposure parameters for the CT scans were: 120 kVp, 100–200 mA, 1–3 mm slice thickness. Image reconstruction for conventional CT scans was performed using the scanner workstation. All CT images were retrieved using a picture archiving and communication system (PACS) (Centricity 4.0; GE Medical Systems, Mountain Prospect, Chicago, USA). Two radiologists (C.H.P. and T.H.K.) with >10 years of experience in chest radiology interpretation assessed the CT images in consensus. We prospectively screened all chest CT images which performed for any causes during study periods in Gangnam Severance hospital. The chest CT was interpreted without any genetic information associated with the BHD syndrome. A lung cyst was defined as an air-filled lesion with a perceptible wall but not more than 3 mm in thickness, and multiple/diffuse cysts were defined as multiple or numerous cysts distributed in both lungs by two experienced radiologists [14]. If the perceptible wall is not visible in the air-filled lesion, it is classified as emphysema, and if the thickness of the lesion wall exceeds 3 mm, it is classified as cavity. The number, size, morphology, and distribution of the lung cysts on chest CT were carefully evaluated. Cases with >40 cysts and those with <40 cysts were categorized separately. Cyst sizes were considered diverse when the coexistence of cysts <2 cm or >2 cm in diameter was observed. Cyst morphology was categorized as round, oval, or irregular, and diverse morphology was defined as the presence of all three types of cyst morphology in one lung. If there was a discrepancy in CT interpretation, a consensus was achieved through a discussion. If a consensus was not achieved for discrepancy, the third independent reader (J.M.S) mediated.

### 2.3. FLCN Gene Mutation Analysis

All enrolled patients underwent DNA testing for *FLCN* gene mutations. Whole blood samples were collected in ethylenediaminetetraacetic acid-containing tubes, and genomic DNA was extracted from each sample using an Easy-DNA™ Kit (Invitrogen, Carlsbad, CA, USA). The concentration and quality of the genomic DNA were assessed using Nanodrop (ND-1000, Thermo Scientific, Wilmington, DE, USA). Primers designed to amplify *FLCN* exons 4–14 and their flanking introns were used for polymerase chain reaction (PCR) amplification. PCR products were purified using a QIA-quick Gel Extraction Kit (Qiagen, Dusseldorf, Germany), and cycle-sequencing was performed using relevant PCR primers and a Big Dye Terminator Cycle Sequencing Ready Reaction Kit (Applied Biosystems, Foster City, CA, USA).

The sequences obtained were compared with the reference sequence using Sequencher^®^ software (Gene Codes, Ann Arbor, MI, USA). Pathogenic variants were detected using Sanger sequencing followed by multiple ligation probe amplification to confirm that there were no large deletions in *FLCN* when no pathogenic variants were detected.

### 2.4. Statistical Analysis

Categorical variables are presented as frequencies (percentages). Continuous variables are presented as mean ± standard deviation (SD) for normally distributed variables and as median (interquartile range [IQR]) for non-normally distributed variables. Normality assumptions for continuous variables were confirmed using the Shapiro–Wilk test. Baseline characteristics were compared between patients with and without BHD using an independent t-test or Mann–Whitney U test. Fisher’s exact test was used to evaluate potential correlations between various chest CT features, using *FLCN* gene mutation results as a reference. Logistic regression analyses were used to evaluate the association between radiological features and BHD. Receiver operating characteristic (ROC) curve analysis was performed to identify the ability of CT features to aid in BHD diagnosis. The area under the curve (AUC) was calculated to assess the sensitivity, specificity, positive predictive value, and negative predictive value of CT findings. The optimal cut-off was defined using Youden’s J statistic [15,16]. A *p* value of 0.05 or less was considered statistically significant.

All statistical analyses were performed using R software version 4.0.2. (R v4.0.2. https://cloud.r-project.org/ (accessed on 31 October 2022)). For ROC curve analysis, the “ROCR” (version 1.0-11) and “pROC” packages (version 1.18.0) were used.

## 3. Results

### 3.1. Baseline Characteristics of Enrolled Patients

Among the 21 enrolled patients, 10 (47%) were diagnosed with BHD based on the presence of *FLCN* gene mutations (Table 1). Eleven exhibited negative results in the *FLCN* gene test. Age, male-to-female ratio, smoking status, comorbidities, proteinuria, haematuria, and PFT did not significantly differ between the two groups. Spontaneous pneumothorax was more frequent in the BHD group (70.0% and 30.0%, respectively) than in the non-BHD group (18.2% and 30.0%, respectively); however, the difference was insignificant (*p* = 0.051 and 0.181, respectively). Similarly, there were no differences in the prevalence of renal cell carcinoma or cutaneous lesions such as fibrofolliculoma or acrochordons between the groups. The prevalence of renal cysts was higher in the BHD group than in the non-BHD group; however, the difference was insignificant (50.0% and 12.5%, respectively; *p* = 0.240).

### 3.2. Clinical Characteristics of BHD

Table 2 shows the detailed clinical characteristics of BHD in each patient. No typical skin or kidney lesions were observed. Forty percent of all BHD patients were assessed for skin lesions by a dermatologist, and non-specific skin lesion (verruca) was observed in one patient (Patient C in Table 2). Renal imaging was available for all BHD patients, half of whom exhibited renal cysts. All BHD patients had normal renal function. In lung function, one patient exhibited impaired lung function with a mild obstructive pattern (Patient F, forced vital capacity [FVC]: 109%, forced expiratory volume in 1 s [FEV1]: 91%, FEV1/FVC: 59%, no response in the bronchodilator test). Seven patients (70.0%) had a personal history of pneumothorax, and three patients had a family history of pneumothorax.

### 3.3. Characteristics of Lung Cysts in BHD Patients

Lung cysts on chest CT exhibited different characteristics in patients with and without BHD (Figure 1). Over 80.0% and 72.7% of patients in the BHD and non-BHD groups exhibited >40 lung cysts, respectively, without significant differences between the groups (*p* = 1.000; Table 3). The maximal diameter of the lung cysts was significantly greater in the BHD group than in the non-BHD group (4.1 ± 1.1 cm vs. 1.6 ± 0.9 cm, *p* < 0.001). Lung cysts were diverse in size and morphology in all BHD cases, and diversity was more prevalent in the BHD group than in the non-BHD group (100% vs. 18.2 %, *p* = 0.001 and 100% vs. 54.5%, *p* = 0.054, respectively).

### 3.4. Correlation between Cystic Features and FLCN Gene Mutations

In the univariate analysis, the maximal diameter of lung cysts was positively correlated with the presence of *FLCN* gene mutations (odds ratio [OR], 95% confidence interval [CI]: 6.352, 2.112–53.223, respectively; Table 4). Contrarily, the number of cysts, diversity in size, and diversity in morphology were not significantly correlated with the presence of *FLCN* mutations (OR [95% CI], 1.500 [0.195–13.860], 999.999 [0.000–999.999], and 999.999 [0.000–999.999], respectively). Similarly, in the multivariate analysis, only the maximal cyst diameter was significantly correlated with the presence of *FLCN* gene mutations (OR [95% CI], 6.884 [1.745–117.748]).

### 3.5. ROC Analysis for BHD Diagnosis

Figure 2 shows the ROC curves for BHD diagnosis. The AUC values for cyst number, maximal diameter, diversity of size, and diversity of morphology were 0.536 (95% CI, 0.346–0.726), 0.955 (95% CI, 0.873–1.000), 0.909 (95% CI, 0.790–1.000), and 0.727 (95% CI, 0.573–0.882), respectively. In the multivariate model combining cyst number, maximal diameter, and morphological diversity, the AUC value was best at 0.964 (0.887–1.000), which was not significantly different from the value observed for maximal diameter alone. The optimum cut-off value for maximal diameter was 2.1 cm, with a sensitivity of 99% and a specificity of 82%.

## 4. Discussion

Characteristics of lung cysts in chest CT are important clues to suspect BHD syndrome and to conduct genetic testing. Classical criteria for BHD based on CT include the presence of multiple bilateral lung cysts that are basally located. In Asia, criteria other than chest CT imaging, including skin manifestations and renal cell cancer, are not prominent [11,12]. Therefore, chest CT findings play an important role for BHD detection in Asia. In addition, our previous prospective study confirmed that classical chest CT criteria only predict 33.3% of BHD cases [11], highlighting the need to identify more detailed chest CT findings that can improve sensitivity and specificity. In the present study, maximal cyst diameter was the single potent radiologic feature of BHD in both the univariate and multivariate logistic regression analyses. Furthermore, maximal cyst diameter exhibited the highest AUC for predicting *FLCN* gene mutation in the ROC analysis.

Previous studies have reported that lung cysts occur in 80–100% of BHD patients, which has been associated with a higher prevalence of lower lung-predominant cysts (100%), elliptical (floppy) paramediastinal cysts (88–94%), and disproportionate number of paramediastinal cysts (69–88%) than other cystic lung diseases [8,17,18,19,20].

In the current study, the maximal diameter of lung cysts was larger in BHD patients than in those with other diseases characterized by multiple lung cysts, and diversity in the size and morphology of lung cysts differed between patients with and without BHD. These results are consistent with those of previous studies [21,22]. It is assumed that these characteristics are associated with the “stretch hypothesis,” although the precise mechanisms underlying this association remain unclear [23]. Maximal cyst diameter was the single potent radiologic feature of BHD in both the univariate and multivariate logistic regression analyses of our study. Additionally, in the ROC curve analysis for predicting *FLCN* gene mutations, the maximal cyst diameter had the highest AUC value, with an optimal cut-off value of 2.1 cm. However, given the small number of patients included in this study and the lack of validation in other cohorts, further research is required to generalize this finding.

The number of lung cysts in BHD patients is reported to vary from 0 to >400 [24,25]. Several studies have reported differences in the number of lung cysts between BHD patients and those with other multiple cystic lung diseases. Tobino et al. [21] reported that the number of lung cysts was smaller in BHD cases than in lymphangioleiomyomatosis cases (28.7 ± 66.9 and 148.6 ± 102.7, respectively, *p* < 0.01). Park et al. [22] reported similar results. However, in this study, the number of lung cysts did not differ between the two groups, which may have been due to the small number of patients and the inability to subdivide the non-BHD group based on specific diagnoses.

There were no significant differences in clinical characteristics between the BHD and non-BHD groups in this study. In non-Asian regions, typical skin lesions are found in >80% of BHD patients. As one of the major criteria of diagnosis for BHD by the European BHD consortium, typical skin lesion (at least 5 fibrofolliculomas or trichodiscomas with onset in adult life and with at least 1 confirmed histopathologically) is a characteristic clinical manifestation that can be diagnosed for BHD without FLCN mutation [8,26]. However, typical skin lesions tend to be less frequently reported in Asian patients than in those of other races, and several studies have demonstrated that only 20% of Asian BHD patients present with typical skin lesions [3,11,27,28]. In the current study, there were also no patients with a typical skin lesion. The reason for the regional difference in skin lesion prevalence may be a racial difference or a bias caused by a retrospective study design, but further research is needed.

Notably, no BHD patients in our study presented with typical renal lesions. In previous studies, renal tumors were observed in approximately 14–34% of BHD patients [29,30]. However, typical renal lesions are also reported less frequently in Asia than in other regions. Yang et al. [31] and Park et al. [11] also reported no typical renal lesions among Asian patients with BHD. Atypical renal cysts were identified in approximately half of the BHD patients in this study; however, as the prevalence of renal cysts in the general population is 27%, renal cysts may not be associated with BHD [32]. Therefore, it suggests that skin and renal lesions in the diagnosis of BHD in Asians will be less important than those in non-Asian people; however, further research is required to evaluate the relationship between race and typical skin or renal lesions in patients with BHD. 

In this study, we observed no significant difference in the history of pneumothorax according to BHD. Previous studies have reported a relationship between a history of pneumothorax and BHD. Houweling et al. [9] reported that 24% of patients carrying *FLCN* gene mutations had a past history of pneumothorax; contrarily, Johannesma et al. [33] reported that primary spontaneous pneumothorax is related to BHD in 10% of cases. We speculate that the insignificance of the difference was due to the small sample size of the current study.

This study had several limitations. First, the number of patients was small; therefore, there is a possibility that statistical significance may not have been achieved. Second, the final diagnoses in patients who were not diagnosed with BHD remained unclear, which may have affected the results. Third, there are technical limitations such as the resolution of chest CT. Recently, CT with higher resolution such as 256-slice MDCT has been used widely. However, in this study, 64-slice MDCT or 128-slice MDCT were used as the main modality. A relatively low-resolution CT technique may have influenced the results. This study also has strengths as a prospective study, as it can provide more accurate insights into the prevalence and characteristics of BHD than retrospective analyses. 

## 5. Conclusions

Chest CT may be useful as an initial screening modality to suggest the possibility of BHD, necessitating the identification of specific radiological criteria for BHD diagnosis. Our findings indicate that reliable chest CT features suggesting the need to screen for *FLCN* gene mutations include variations in cyst size and the presence of cysts >2.1 cm in diameter, predominantly occurring in the bilateral basal lung zones.

## Figures and Tables

**Figure 1 diagnostics-13-00198-f001:**
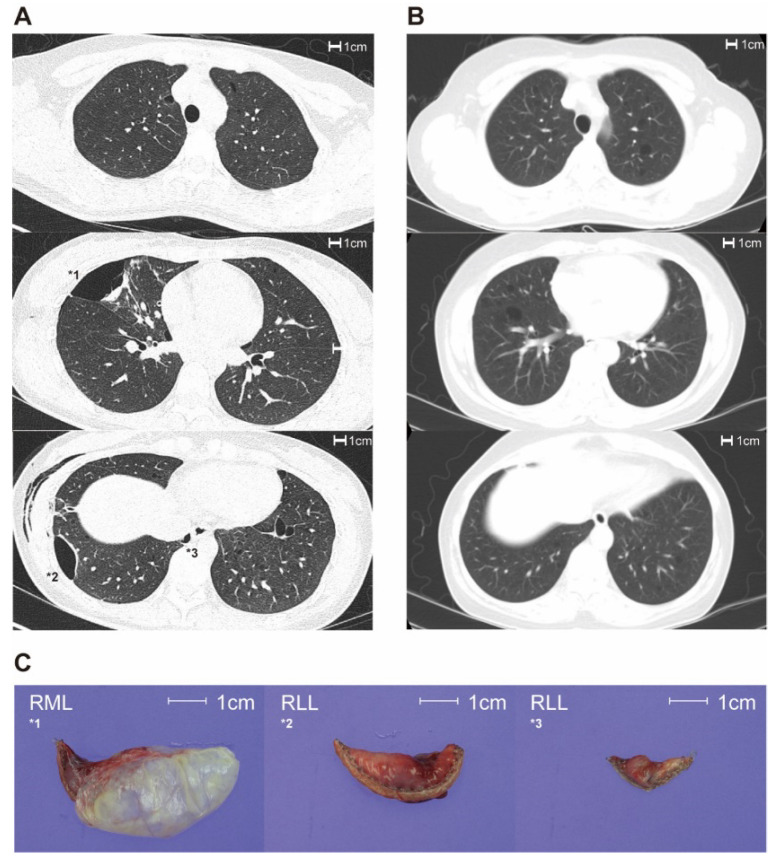
Differences in the characteristics of lung cysts between the BHD and non-BHD groups. (**A**): Transverse plane view of chest computed tomography for BHD patients. (**B**): Transverse plane view of chest computed tomography for non-BHD. (**C**): Gross pathology images of lung cysts in BHD patient, The number of cyst is same as marked number in chest computed tomography image (**A**). BHD: Birt–Hogg–Dube syndrome.

**Figure 2 diagnostics-13-00198-f002:**
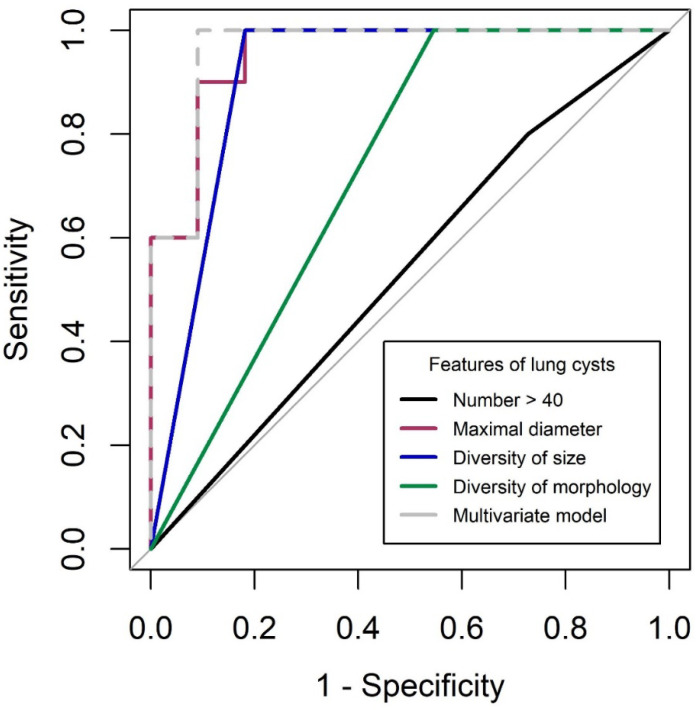
Receiver operating characteristic analysis for diagnosis of BHD. BHD: Birt–Hogg–Dube syndrome.

**Table 1 diagnostics-13-00198-t001:** Baseline characteristics of patients according to the results of screening for *FLCN* gene mutations.

	BHD	Non-BHD	*p*-Value
	(N = 10)	(N = 11)	
Sex			0.730
Male	2 (20.0%)	4 (36.4%)	
Female	8 (80.0%)	7 (63.6%)	
Age (years; mean ± standard deviation)	49.4 ± 16.2	50.4 ± 12.4	0.879
Smoking status			0.961
Never smoker	9 (90.0%)	11 (100.0%)	
Current smoker	1 (10.0%)	0 (0.0%)	
Comorbidities			
Hypertension	3 (30.0%)	0 (0.0%)	0.181
Diabetes	0 (0.0%)	1 (9.1%)	1.000
Tuberculosis	0 (0.0%)	0 (0.0%)	
Proteinuria	0 (0.0%)	0 (0.0%)	
Hematuria	2 (28.6%)	1 (10.0%)	0.732
Pulmonary function test			
FVC (%; mean ± standard deviation)	94.8 ± 11.5	91.8 ± 10.1	0.603
FEV1 (%; median [IQR])	91.0 [9]	93.0 [17]	0.796
FEV1/FVC (%; mean ± standard deviation)	72.8 ± 7.8	79.5 ± 16.7	0.384
History of spontaneous pneumothorax	7 (70.0%)	2 (18.2%)	0.051
Family history of spontaneous pneumothorax	3 (30.0%)	0 (0.0%)	0.181
Typical skin lesions †	0 (0.0%)	0 (0.0%)	NA
Typical renal lesions ‡	0 (0.0%)	0 (0.0%)	NA
Renal cyst	5 (50.0%)	1 (12.5%)	0.240

BHD, Birt–Hogg–Dube syndrome; FVC, forced vital capacity; FEV1, forced expiratory volume in 1 s; IQR, interquartile range; NA, not available; † Fibrofolliculomas or trichodiscomas; ‡ Early onset renal cancer (<50 years), multifocal or bilateral location, or mixed chromophobe and oncocytic histology.

**Table 2 diagnostics-13-00198-t002:** Clinical and *FLCN* gene characteristics of BHD patients.

Patient	Age	Sex	Skin Lesion	Renal Lesion	Renal Function Impairment	Lung Function Impairment	History of Pneumothorax		*FLCN* Gene Mutation
							Personal History	Family History ‡	
A	50	M	None	Cortical single cyst †, 1.1 cm	None	None	Positive	Negative	c.1285dupC
B	31	F	None	None	None	None	Positive	Positive	c.1177-5_1177-3delCTC
C	56	F	Verruca †	Cortical multiple cyst †, tiny	None	None	Negative	Negative	c.1023_1024insTCTTC
D	54	F	None	Cortical single cyst †, 2.1 cm	None	None	Positive	Negative	c.1177-5_1177-3delCTC
E	53	F	None	Cortical multiple cyst †, tiny	None	N/A	Positive	Positive	c.507G>A
F	70	F	None	Medullary single cyst †, 0.9 cm	None	Obstructive	Negative	Negative	c.1128G>A
G	35	M	None	None	None	N/A	Positive	Negative	c.469_471delTTC
H	24	F	None	None	None	N/A	Positive	Negative	c.1285delC
I	46	F	None	None	None	None	Negative	Positive	c.1285dupC
J	75	F	None	None	None	N/A	Positive	Negative	c.469_471delTTC

BHD, Birt–Hogg–Dube syndrome; † Atypical features of BHD; ‡ Family history within second-degree relatives.

**Table 3 diagnostics-13-00198-t003:** Characteristic chest CT findings according to *FLCN* gene mutation status.

Features of Lung Cysts	BHD	Non-BHD	*p*-Value
	(N = 10)	(N = 11)	
Number > 40	8 (80%)	8 (72.7%)	1.000
Maximal diameter (Centimeters; mean ± standard deviation)	4.1 ± 1.1	1.6 ± 0.9	<0.001 **
Diversity of size	10 (100%)	2 (18.2%)	0.001 *
Diversity of morphology	10 (100%)	6 (54.5%)	0.054

BHD, Birt–Hogg–Dube syndrome; CT, computed tomography; * *p* < 0.05, ** *p* < 0.001.

**Table 4 diagnostics-13-00198-t004:** Logistic regression between radiologic features and BHD.

Features of Lung Cysts	Univariate Analysis		Multivariate Analysis †	
	Odds Ratio (95% Confidence Interval)	*p*-Value	Odds Ratio (95% Confidence Interval)	*p*-Value
Number > 40	1.500 (0.195–13.860)	0.697	1.306 (0.001–6.392)	0.341
Maximum diameter (cm)	6.352 (2.112–53.223)	0.014	6.884 (1.745–117.748)	0.048 *
Diversity of size	999.999 (0.000–999.999)	0.995	Omitted	
Diversity of morphology	999.999 (0.000–999.999)	0.995	999.999 (0.000–999.999)	0.997

AUC, area under the curve; ROC curve, receiver operating characteristic curve; BHD, Birt–Hogg–Dube syndrome; CT, computed tomography; IQR, interquartile range; * < 0.05; † Diversity of size was omitted from the multivariate analysis because it was highly correlated with the maximum diameter.

## Data Availability

The data that support the findings of this study are available on request from the corresponding author. The data are not publicly available due to privacy or ethical restrictions.

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
