# Peer review of "Characteristic Chest Computed Tomography Findings for Birt–Hogg–Dube Syndrome Indicating Requirement for Genetic Evaluation"

_diagnostics, 2023, doi:10.3390/diagnostics13020198_

Round 1

Reviewer 1 Report

This single-center prospective observational study investigated the radiologic features of chest computed tomography (CT) of patients with Birt–Hogg–Dube (BHD) Syndrome. In this study, compared to cystic lung diseases caused by non-BHD, lung cysts of the BHD patients showed a larger maximal diameter (4.1±1.1 vs. 1.6±0.9, P<0.001). In the multivariate analysis of correlation between cystic features and FLCN gene mutations, maximal cyst diameter was significantly correlated with the presence of FLCN gene mutations (OR [95% CI], 6.884 [1.745–117.748]). Although this is an interesting piece of work, there are several issues to be clarified.

1.      The detection rate of an FLCN gene mutation was approximately 84% to 88% in BHD syndrome. (References – “Germline BHD-mutation spectrum and phenotype analysis of a large cohort of families with Birt-Hogg-Dubé syndrome. Am J Hum Genet 2005;76:1023–1033” and “BHD mutations, clinical and molecular genetic investigations of Birt-Hogg-Dubé syndrome: a new series of 50 families and a review of published reports. J Med Genet 2008;45:321–331”) Therefore, non-BHD group could not be excluded for the diagnosis of BHD only based on the absence of FLCN gene mutations. Please address the criteria proposed by the European BHD consortium in your manuscript.

2.      Please specify how enrolled patients with multiple lung cysts were included in details (for example, text search of CT reading and the detailed process, criteria of multiple cysts).

Author Response

Point 1: The detection rate of an FLCN gene mutation was approximately 84% to 88% in BHD syndrome. (References – “Germline BHD-mutation spectrum and phenotype analysis of a large cohort of families with Birt-Hogg-Dubé syndrome. Am J Hum Genet 2005;76:1023–1033” and “BHD mutations, clinical and molecular genetic investigations of Birt-Hogg-Dubé syndrome: a new series of 50 families and a review of published reports. J Med Genet 2008;45:321–331”) Therefore, non-BHD group could not be excluded for the diagnosis of BHD only based on the absence of FLCN gene mutations. Please address the criteria proposed by the European BHD consortium in your manuscript.

Response 1: We highly appreciate for your valuable comment. As you pointed out, European Birt-Hogg-Dubé syndrome (BHD) consortium suggests diagnosis criteria of BHDS which included 2 major criteria; 1) at least 5 fibrofolliculomas or trichodiscomas with onset in adult life and with at least 1 confirmed histopathologically, 2) a pathogenic FLCN germline mutation; and 3 minor criteria; 1) multiple lung cysts: bilateral basal cysts with no other apparent cause, with or without spontaneous pneumothorax, 2) multifocal or bilateral renal cancer of early onset (>50 years), or renal cancer of mixed chromophobe and oncoytic histology, 3) first-degree relative with Birt-Hogg-Dubé syndrome. Diagnosis of Birt-Hogg-Dubé syndrome requires the presence of 1 major or 2 minor criteria by this diagnostic criteria. Therefore, BHD diagnosis can be made without FLCN germline mutation by major criteria 1) with 2 minor criteria. However, all non-BHD patients in our research did not present any specific skin lesions (Table 1), therefore the FLCN germline mutation was the only criterion which is included in major criteria in this setting. For this reason, non-BHD group could be excluded for the diagnosis of BHD only based on the absence of FLCN gene mutations. Your comment is a very important part for the diagnosis of BHD, we added the following sentences to the discussion section.

DISCUSSION

“As one of the major criteria of diagnosis for BHD by the European BHD consortium, typical skin lesion (at least 5 fibrofolliculomas or trichodiscomas with onset in adult life and with at least 1 confirmed histopathologically) is a characteristic clinical manifestation that can be diagnosed for BHD without FLCN mutation.[8,26]”

Point 2: Please specify how enrolled patients with multiple lung cysts were included in details (for example, text search of CT reading and the detailed process, criteria of multiple cysts).

Response 2: We appreciate your valuable comment. We prospectively screened all chest CT images which performed during study pertiods by two radiologists with more than ten years of experience. Chest CT images were interpreted before the genetic analysis. Patients who are satisfied with inclusion criteria in chest CT images were reported to the clinicians, we explained this study to them and enrolled only patients who agreed to participate. Lung cyst was defined as an air-filled lesion with a perceptible wall but not more than 3 mm in thickness by two experienced radiologists. Single or several cysts in a localized area are classified as solitary/localized cysts, while multiple or numerous cysts distributed in both lungs are classified as multiple/diffuse cysts. We added these contents in the method section.

METHODS, 2.3. CT protocol and analysis

“We prospectively screened all chest CT images which performed for any causes during study pertiods in Gangnam Severance hospital. The chest CT was interpretated without any genetic information associated with the BHD syndrome. Lung cyst was defined as an air-filled lesion with a perceptible wall but not more than 3 mm in thickness, and multi-ple/diffuse cysts defined as multiple or numerous cysts distributed in both lungs by two experienced radiologists.[14] If the perceptible wall is not visible in the air-filled lesion, it is classified as empysema, and if the thickness of the lesion wall exceeds 3mm, it is classi-fied as cavity.”

Reviewer 2 Report

I would like to thank the handling editor for giving me the opportunity to review the manuscript entitled “Characteristic chest computed tomography findings for Birt-Hogg-Dube syndrome indicating requirement for genetic evaluation” by Choi and colleagues. This is a single-centre, prospective study investigating radiographic characteristics of Birt-Hogg-Dube syndrome found on chest computed tomography that predict the need for genetic examination for FLCN mutations. The study included 21 patients with multiple bilateral and basally located lung cysts on chest computed tomography with no other apparent cause, of whom 10 were diagnosed with Birt-Hogg-Dube syndrome. The authors found that maximal cyst diameter was significantly greater in the Birt-Hogg-Dube syndrome group (4.1 cm ± 1.1 cm) than the control group (1.6 cm ± 0.9 cm; P<0.001). Similarly, diversity in cyst size was observed in all patients with Birt-Hogg-Dube syndrome, as compared to 18.2% of the patients in the control group (P=0.001). Furthermore, areas under the receiver operating characteristic curves for predicting FLCN gene mutations were 0.955 and 0.909 for maximal cyst diameter and diversity in size, respectively. Therefore, the authors conclude that reliable chest computed-tomography features suggesting the need for FLCN gene mutations screening include variations in cyst size and presence of cysts greater than 2.1 cm in diameter, predominantly occurring in the bilateral basal lungs. This is an interesting and informative study that can be a valuable addition to the relevant literature.

The paper is overall well-written. The introduction sets the appropriate background even for the reader with little knowledge on the topic. The results are clearly presented with relevant tables and figures. The findings are discussed within the context of the pertinent literature, and the conclusions are based on the results of the study. Here, I have made a few suggestions that (in my opinion) could help improve the overall quality of the manuscript.

·         The authors may consider clarifying a few aspects of their methodology. They may state the reason why the study population underwent computed tomography (e.g., screening program for certain diseases, inpatient or outpatient investigations, etc.) and if these patients were consecutive or not. They may also report if the radiologists were aware of the results of the genetic analysis at the time of assessment of the computed tomography scans.

·         The authors may consider reporting what the diagnoses were in the control group. If such diagnoses were not established, they may consider reporting if these patients were further investigated for other potential cystic lung diseases and what investigations they underwent.

·         The authors may consider reporting data on the thickness of the cyst wall in patients with Birt-Hogg-Dube syndrome.

·         It has been previously reported that the majority of cysts in patients with Birt-Hogg-Dube syndrome are located in the basilar medial regions of the lungs. The authors may consider reporting the basilar segments in which cysts were predominantly observed in the study patients.

Author Response

Point 1: The authors may consider clarifying a few aspects of their methodology. They may state the reason why the study population underwent computed tomography (e.g., screening program for certain diseases, inpatient or outpatient investigations, etc.) and if these patients were consecutive or not. They may also report if the radiologists were aware of the results of the genetic analysis at the time of assessment of the computed tomography scans.

Response 1: We highly appreciate your valuable comment. In our study, we prospectively screened all chest CT images which performed for any causes during study pertiods in Gangnam Severance hospital. The chest CT was interpretated without any genetic informations associated with the BHD syndrome. In our process, Patients’ CT images are compatible with inclusion criteria of BHD Chest CT scans were reported to the clinicians, we explained this study to patients, enrolled patients who agreed to participate, and performed the DNA testing for FLCN gene mutations. To clarify this aspect, we added following sentences in the method section.

METHODS, 2.3. CT protocol and analysis

“We prospectively screened all chest CT images which performed for any causes during study pertiods in Gangnam Severance hospital. The chest CT was interpretated without any genetic information associated with the BHD syndrome. Lung cyst was defined as an air-filled lesion with a perceptible wall but not more than 3 mm in thickness, and multi-ple/diffuse cysts defined as multiple or numerous cysts distributed in both lungs by two experienced radiologists.[14]”

Point 2: The authors may consider reporting what the diagnoses were in the control group. If such diagnoses were not established, they may consider reporting if these patients were further investigated for other potential cystic lung diseases and what investigations they underwent.

Response 2: We appreciate your reasonable comment. Because a variety of pathophysiological processes and diseases can present as lung cysts and cystic lung diseases, it is important to identify diseases that can cause multiple lung cysts. However, pathological or genetic evaluation for diagnosis of other diseases was not performed in our study design. Although patients with other diseases such as lymphangioleiomyomatosis, pulmonary Langerhans cell histiocytosis, and lymphocytic interstitial pneumonia were excluded in the screening stage, diseases that could cause multiple cysts were not identified in non-BHD patients. Therefore, we mentioned this point in the discussion section as a limitation of our study. And to overcome these limitations, our subsequent studies have tried to detect other possible genetic abnormalities (TSC1, TSC2, CFTR, ELN, LTBP4, and SERPINA1) in addition to FLCN. Thank you again for your valuable opinion.

Point 3: The authors may consider reporting data on the thickness of the cyst wall in patients with Birt-Hogg-Dube syndrome.

Response 3: Thank you for your valid and considerable points to improve our manuscript. We had a deep discussion about whether thickness measurement is possible as your comment. When we evaluated chest CT, cyst was defined as an air-filled lesion with a perceptible wall but not more than 3mm in thickness, and was evaluated by two experimental radiologists. In addition, if the perceptible wall is not visible in the air-filled lesion, it is classified as empysema, and if the thickness of the lesion wall exceeds 3mm, it is classified as cavity. Therefore, the thickness of each ‘lung cyst’ is perceptible, but it is not more than 3mm. In conclusion, quantitatively measuring the wall thickness of lung cyst lesion is difficult considering measurement errors. However, we believe that thickness of cyst can be one of the meaningful variables, so we will devise a reasonable thickness measurement method so that this measurement can proceed in the next study.

Point 4: It has been previously reported that the majority of cysts in patients with Birt-Hogg-Dube syndrome are located in the basilar medial regions of the lungs. The authors may consider reporting the basilar segments in which cysts were predominantly observed in the study patients.

Response 4: As you commented, the location and characteristics of cysts in patients with BHD provide important clues for differential diagnosis from other diseases. Menko et al. and Gupta et al. reported that locations of the cyst is characterized by a predominant distribution at basilar/peripheral/ subpleural (PMID 19959076 and PMID 23715758, respectively). We also reflected these characteristics in the inclusion criteria. Therefore, multiple bilateral and basal cysts were shown in both BHD and non-BHD groups in our study, and were not reported as a specific variable. However, since it is an important feature, we described it as follows in the method section.

METHODS, 2.2. Patients

“From June 2016 to December 2017, we screened all patients who underwent chest CT and prospectively enrolled patients with multiple bilateral and basally located lung cysts on chest CT, which are minor criteria for the diagnosis of BHD in previous studies.[7,13]

Reviewer 3 Report

Dear Editor and Authors,

Thank you for asking me to review this work for Diagnostics titled “Characteristic Chest Computed Tomography Findings for Birt-Hogg–Dube Syndrome Indicating Requirement for Genetic Evaluation” by Dr. Yong Jun Choi and colleagues from Seoul, Korea.

In this study the authors attempt to elucidate more specific and characteristic CT findings for  Birt-Hogg–Dube Syndrome which can be used as a screening method to identify patients needing to undergo folliculin gene (FLCN) mutations testing.

After prospectively enrolling 21 patients which could fit the profile of the syndrome (multiple bilateral and basally located lung cysts with no apparent causative mechanism) and following genetic testing they were able to identify ten patients with Birt-Hogg–Dube Syndrome. Their radiological findings were analyzed to look for characteristic and common patterns. They were able to demonstrate that maximal cyst diameter and diversity in size could be used to identify genetic mutation.

This is overall a small and simple study from a single institution. It has clear and concise methodology with well-established parameters.   Given the rarity of the condition the study has a small sample of patients and this is quite rightly mentioned as a limitation by the authors. Otherwise the manuscript is well written, clear to understand in good language and it is well illustrated and presented.

I do have a few questions/queries that I would like clarification on:

1.       In term of the inclusion criteria in the study (line 73) I wonder if they should mention mild COPD and chronic emphysematous disease as a causative mechanism for lung cysts which was excluded. In truth their disease pattern is quite distinctive compared to that of the other symptoms mentioned but just for clarification purposes for the reader a short sentence is needed.

2.       Given the progress in CT scanning technology why was a 64 slice MDCT scanner used and not a more higher resolution one (I note that  a 128 slice was also used but now we are at the area of 256 slice CTs)!! The authors need to mention this as a limitation to their study as it reduces the information they could have drawn!!

3.       Was assessment of findings performed independently or together by the two radiologists? If not independently which is the methodological correct approach why was it performed together? If there was a discrepancy between the two radiologists assessment (and no consensus) how was that resolved? Was a third independent assessor utilized?

In conclusion, this is a small but quite well conducted and reported study on which I only have minor commentary.  The condition it assesses is rare so it has inherent interest. I would be happy to recommend its publication when my quite few questions have been answered.

I wish all well and kind regards.

Author Response

Point 1: In term of the inclusion criteria in the study (line 73) I wonder if they should mention mild COPD and chronic emphysematous disease as a causative mechanism for lung cysts which was excluded. In truth their disease pattern is quite distinctive compared to that of the other symptoms mentioned but just for clarification purposes for the reader a short sentence is needed.

Response 1: We highly appreciated for your thoughtful comment. As we described in the exclusion criteria, patients with multiple cysts caused by other apparent causes were excluded. The apparent cause included lymphangioleiomyomatosis, pulmonary Langerhans cell histiocytosis, lymphocytic interstitial pneumonia, and chronic obstructive pulmonary disease. In addition, cyst was defined as an air-filled lesion with a perceptible wall but not more than 3mm in thickness and was evaluated by two experimental radiologists. If the perceptible wall is not visible in the air-filled lesion, it is classified as emphysema, and if the thickness of the lesion wall exceeds 3mm, it is classified as cavity. In this process, COPD patients were excluded through past history and CT findings. To clarify these things, we added as follows.

METHODS, 2.2. Patients

“Patients exhibiting cysts with other apparent causes were excluded such as chronic obstructive pulmonary disease, lymphangioleiomyomatosis, pulmonary Langerhans cell histiocytosis, and lymphocytic interstitial pneumonia, although cases both with and without spontaneous primary pneumothorax were included.”

METHODS, 2.3. CT protocol and analysis

“Lung cyst was defined as an air-filled lesion with a perceptible wall but not more than 3 mm in thickness, and multiple/diffuse cysts defined as multiple or numerous cysts dis-tributed in both lungs by two experienced radiologists.[14] If the perceptible wall is not visible in the air-filled lesion, it is classified as emphysema, and if the thickness of the le-sion wall exceeds 3mm, it is classified as cavity.”

Point 2: Given the progress in CT scanning technology why was a 64 slice MDCT scanner used and not a more higher resolution one (I note that a 128 slice was also used but now we are at the area of 256 slice CTs)!! The authors need to mention this as a limitation to their study as it reduces the information they could have drawn!!

Response 2: As you point out, CT scanning technology is one of the obvious limitations of our study. Unfortunately, the upgrade of CT device has been relatively recent in our center. Therefore, from 2016 to 2017, when this study was conducted, the main modality of chest CT in our center was 64 slice or 128 slice MDCT. This limitation was added to discussion section as follows.

DISCUSSION

“Third, there are technical limitations such as resolution of chest CT. Recently, CT with higher resolution such as 256-slice MDCT have been used widely. However, in this study, 64-slice MDCT or 128-slice MDCT were used as main modality. A relatively low-resolution CT technique may have influenced the results.”

Point 3: Was assessment of findings performed independently or together by the two radiologists? If not independently which is the methodological correct approach why was it performed together? If there was a discrepancy between the two radiologists assessment (and no consensus) how was that resolved? Was a third independent assessor utilized?

Response 3:

Thank you for a valid comment. In our study design, two experienced radiologists evaluated the radiological imaging independently, after which if there was discrepancy, a consensus was achieved through a discussion. If a consensus was not achieved for discrepancy, the third independent reader mediated. This is an important methodology for CT image evaluation; therefore, the following sentences were added to the method section.

METHODS, 2.3. CT protocol and analysis

“If there was a discrepancy in CT interpretation, a consensus was achieved through a discussion. If a consensus was not achieved for discrepancy, the third independent reader (J.M.S) mediated.”

Round 2

Reviewer 2 Report

Thank you for considering my suggestions and revising your manuscript accordingly.